# Identifying Bias in AI using Simulation

## Abstract

Machine learned models exhibit bias, often because the datasets used to train them are biased. This presents a serious problem for the deployment of such technology, as the resulting models might perform poorly on populations that are minorities within the training set and ultimately present higher risks to them. We propose to use high-fidelity computer simulations to interrogate and diagnose biases within ML classifiers. We present a framework that leverages Bayesian parameter search to efficiently characterize the high dimensional feature space and more quickly identify weakness in performance. We apply our approach to an example domain, face detection, and show that it can be used to help identify demographic biases in commercial face application programming interfaces (APIs).

## 1 Introduction

Machine learned classifiers are becoming increasingly prevalent and important. Many systems contain components that leverage trained models for detecting or classifying patterns in data. Whether decisions are made entirely, or partially based on the output of these models, and regardless of the number of other components in the system, it is vital that their characteristics are well understood. However, the reality is that with many complex systems, such as deep neural networks, many of the "unknowns" are unknown and need to identified (Lakkaraju et al., 2016; 2017). Imagine a model being deployed in law enforcement for facial recognition, such a system could encounter almost infinite scenarios; which of these scenarios will the classifier have a blind-spot for? We propose an approach for helping diagnose biases within such a system more efficiently.

Many learned models exhibit bias as training datasets are limited in size and diversity (Torralba & Efros, 2011; Tommasi et al., 2017), or they reflect inherent human-biases (Caliskan et al., 2017). It is difficult for researchers to collect vast datasets that feature equal representations of every key property. Collecting large corpora of training examples requires time, is often costly and is logistically challenging. Let us take facial analysis as an exemplar problem for computer vision systems. There are numerous companies that provide services of face detection and tracking, face recognition, facial attribute detection, and facial expression/action unit recognition (e.g., Microsoft (msf, 2018), Google (goo, 2018), Affectiva (McDuff et al., 2016; aff, 2018)). However, studies have revealed systematic biases in results of these systems (Buolamwini, 2017; Buolamwini & Gebru, 2018), with the error rate up to seven times larger on women than men. Such biases in performance are very problematic when deploying these algorithms in the real-world. Other studies have found that face recognition systems misidentify [color, gender (women), and age (younger)] at higher error rates (Klare et al., 2012). Reduced performance of a classifier on minority groups can lead to both greater numbers of false positives (in a law enforcement domain this would lead to more frequent targeting) or greater numbers of false negatives (in a medical domain this would lead to missed diagnoses).

Taking face detection as a specific example of a task that all the services mentioned above rely upon, demographic and environmental factors (e.g., gender, skin type, ethnicity, illumination) all influence the appearance of the face. Say we collected a large dataset of positive and negative examples of faces within images. Regardless of how large the dataset is, these examples may not be evenly distributed across each demographic group. This might mean that the resulting classifier performs much less accurately on African-American people, because the training data featured few examples. A longitudinal study of police departments revealed that African-American individuals were more likely to be subject to face recognition searches than others (Garvie, 2016). To further complicate matters, even if one were to collect a dataset that balances the number of people with different skin types, it is highly unlikely that these examples would have similar characteristics across all

other dimensions, such as lighting, position, pose, etc. Therefore, even the best efforts to collect balanced datasets are still likely to be flawed. The challenge then is to find a way of successfully characterizing the performance of the resulting classifier across all these dimensions.

The concept of *fairness through awareness* was presented by Dwork et al. (2012), the principle being that in order to combat bias we need to be aware of the biases and why they occur. This idea has partly inspired proposals of standards for characterizing training datasets that inform consumers of their properties (Gebru et al., 2018; Holland et al., 2018). Such standards would be very valuable. However, while transparency is very important, it will not solve the fundamental problem of how to address the biases caused by poor representation. Nor will it help identify biases that might still occur even with models trained using carefully curated datasets.

Attempts have been made to improve facial attribute detection by including gender and racial diversity. In one example, by Ryu et al. (2017), results were improved by scraping images from the web and learning facial representations from a held-out dataset with a uniform distribution across race and gender intersections. However, a drawback of this approach is that even images available from vast sources, such as Internet image search, may not be evenly balanced across all attributes and properties and the data collection and cleaning is still very time consuming.

To address the problem of diagnosing bias in real world datasets we propose the use of high-fidelity simulations (Shah et al., 2018) to interrogate models. Simulations allow for large volumes of diverse training examples to be generated and different parameter combinations to be systematically tested, something that is challenging with "found" data scrapped from the web or even curated datasets.

Simulated data can be created in different ways. Generative adversarial networks (GANs) (Goodfellow et al., 2014) are becoming increasingly popular for synthesizing data (Shrivastava et al., 2017). For example, GANs could be used to synthesize images of faces at different ages (Yang et al., 2017). However, GANs are inherently statistical models and are likely to contain some of the biases that the data used to train them contain. A GAN model trained with only a few examples of faces with darker skin tones will likely fail to produce a diverse set of high quality synthesized images with this attribute. Parameterized graphics models are an alternative for training and testing vision models (Veeravasarapu et al., 2015a;b; 2016; Vazquez et al., 2014). Specifically, it has been proposed that graphics models be used for performance evaluation (Haralick, 1992). As an example, this approach has been used for models for pedestrian detection (Vazquez et al., 2014). To the best of our knowledge graphics models have not been employed for detecting demographic biases within vision models. We believe that demographic biases in machine learned systems is significant enough a problem to warrant further attention.

The contributions of this paper are to: (1) present a simulated model for generating synthetic facial data, (2) show how simulated data can be used to identify the limitations of existing face detection algorithms, and (3) to present a sample efficient approach that reduces the number of simulations required. The simulated model used in this paper is made available.

## 2 RELATED WORK

### 2.1 BIAS IN MACHINE LEARNING

With more adoption of machine learned algorithms in real-world applications there is growing concern in society that these systems could discriminate between people unfairly[1][2]. In cases where these systems provide a reliable signal that the output has a low confidence, these can be described as *known unknowns*. However, many learned models are being deployed while their performance in specific scenarios is still unknown, or their prediction confidence in scenarios is not well characterized. These can be described as *unknown unknowns*.

Lakkaraju et al. (Lakkaraju et al., 2016; 2017) published the first example of a method to help address the discovery of unknowns in predictive models. First, the search-space is partitioned into groups which can be given interpretable descriptions. Second, an explore-exploit strategy is used to navigate through these groups systematically based on the feedback from an oracle (e.g., a human

---

[1] https://www.newscientist.com/article/2166207-discriminating-algorithms-5-times-ai-showed-prejudice/
[2] https://www.technologyreview.com/s/608986/forget-killer-robotsbias-is-the-real-ai-danger/

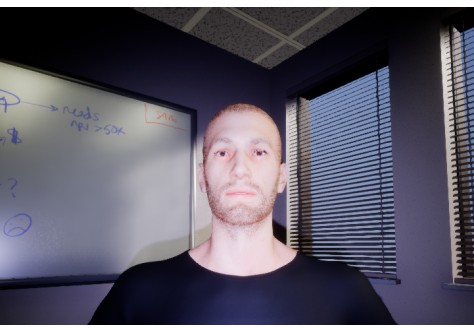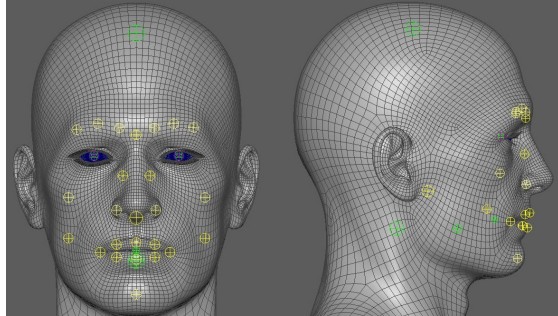

Figure 1: Our simulated environment and the bone positions on the agent model. The head pose, facial expressions, skin type, age, camera position and lighting are all fully customizable.

labeler). Bansal and Weld proposed a new class of utility models that rewarded how well the discovered *unknown unknowns* help explain a sample distribution of expected queries (Bansal & Weld, 2018). We also employ an explore-exploit strategy in our work, but rather than rely on an oracle we leverage synthetic data.

Biases often result from unknowns within a system. Biases have been identified in real-world automated systems applied in domains from medicine (Zheng et al., 2017) to criminal justice (Angwin et al., 2016). While the exact nature of the problem is debated (Flores et al., 2016) it is clear that further research is needed to address bias in machine learned systems and actionable remedies to help practitioners in their day to day work are necessary.

Numerous papers have highlighted biases within data typically used for machine learning (Caliskan et al., 2017; Torralba & Efros, 2011; Tommasi et al., 2017) and machine learned classifiers (Buolamwini & Gebru, 2018). Biases in datasets can be caused in several ways. Torralba and Efros (2011) identified selection bias, capture bias, and negative set bias as problems. "Selection bias" is related to the tendency for certain types, or classes, of images to be included in datasets in the first place. "Capture bias" is related to how the data are acquired and may be influenced by the collectors' preferences (e.g. in images of certain people, say celebrities, or points of view, lighting conditions, etc.). Taking faces as an example, these biases typically manifest as a greater proportion of images including white males, frontal head poses and smiles or neutral expressions. "Negative set bias" is related to the examples in the dataset of the "rest of the world". As an example, if the negative examples included in a dataset of faces are skewed, the model will learn a warped representation of what is not a face.

However, even if one were to make every effort to address these three types of bias, the problem may still exist that artifacts created by humans contain the biases that human have without our explicit knowledge (Caliskan et al., 2017). In our discussion, we will call this "human bias".

With increasing frequency, data hungry machine learning classifiers are being trained on large-scale corpora collected from Internet sources (Deng et al., 2009; Fabian Benitez-Quiroz et al., 2016)) and/or via crowdsourcing (McDuff et al., 2018). Such data collection makes curation challenging as there are a vast number of samples to label and sort. The Internet is a data source that is likely to be subject to "selection bias", "capture bias", "negative set bias" and "human bias".

Furthermore, Tommasi et al. (2017) argue that more powerful feature descriptors (such as those generated by a deep convolutional neural network versus a simple hand crafted approach) may actually exacerbate the problem of bias and accentuate biases in the resulting classifier. Even in the best case scenario, feature representations alone cannot intrinsically remove the negative bias problem.

Thus there is still a need for approaches that help address the issues of bias within machine learning, and this problem is becoming more acute. To this end, we propose a practical approach to use high-fidelity simulations to diagnoses biases efficiently in machine vision classifiers.

## 2.2 FACE DETECTION

Face detection was one of the earliest applications of computer vision. Below we describe significant landmarks in the development of face detection algorithms. Earlier methods relied on rigid templates with boosting learning methods commonly employed. Hand-crafted features used for learning included Haar like features, local binary patterns and histograms of gradients. A major landmark was the Viola-Jones (Viola & Jones, 2004) classifier based on Haar features. This work presented a fast and accurate method and was widely used thanks to implementations in OpenCV and other computing frameworks.

Deep convolutional neural networks have since surpassed the performance of previous methods (Farfade et al., 2015; Yang et al., 2015) and can be used to learn face detection and attribute classification together in a single framework (Ranjan et al., 2017). Face detection has matured into a technology that companies and agencies are deploying in many real world contexts (Zafeiriou et al., 2015); these include safety critical applications. Application programming interfaces (APIs) and software development kits (SDKs) are two ways in which companies are exposing these models to other businesses and to consumers. All the APIs we study in this paper use convolutional neural networks for face detection, making diagnosing points of failure or bias challenging. The approach we present for identifying biases is algorithm independent, but we believe is particularly suited to models trained on large corpra and using powerful feature descriptors.

Face detection is frequently the first step applied in screening images to include in datasets used for subsequent facial analysis tasks (e.g., face recognition or expression detection). If the face detector used for such tasks is biased, then the resulting data set is also likely to be biased. This was found to be the case with the commonly used Labeled Faces in the Wild (LFW) dataset (Huang & Learned-Miller, 2014). One study found it to be 78% male and 85% White (Han & Jain, 2014). We believe a simulation-based approach, using an artificial human (see Figure 1), could help characterize such biases and allow researchers to account for them in the datasets they collect.

## 3 APPROACH

We propose to use simulation to help interrogate machine learned classifiers and diagnose biases. The key idea here is that we repeatedly synthesize examples via simulation that have the highest likelihood of breaking the learned model. By synthesizing such a set, we can then take a detailed look at the failed examples in order to understand the biases and the blind spots that were baked into the model in the first place. We leverage the considerable advancements in computer graphics to simulate highly realistic inputs. In many cases, face detection algorithms are designed to be able to detect faces that occupy as few as 36 x 36 pixels. The appearances of the computer generated faces are as close to photo-realistic in appearance as is necessary to test the face detectors, and at low resolutions we would argue that they are indistinguishable from photographs.

While simulated environments allow us to explore a large number of parameter combinations quickly and systematically, varying lighting, head pose, skin type, etc. often require significant computational resources. Furthermore, exhaustive search over the set of simulations is rendered infeasible as each degree of freedom in simulation leads to exponentially increasing numbers of examples. Therefore, we need a learning method that can intelligently explore this parameter space in order to identify regions of high failure rates.

In this work, we propose to apply Bayesian Optimization (Brochu et al., 2010) to perform this parameter search efficiently. Formally, lets denote $\theta$ as parameters that spawns an instance of simulation $s(\theta)$. This instance then is fed into the machine intelligence system in order to check whether the system correctly identifies $s(\theta)$. Consequently, we can define a composite function $\mathrm{Loss}(s(\theta))$, that captures the notion of how well the AI system handles the simulation instance generated when applying the parameters $\theta$. Note that the function $\mathrm{Loss}(\cdot)$ is similar to the loss functions used in training classifiers (e.g. 0-1 loss, hinge loss etc.). Our goal then is to find diverse instances of $\theta$ such that the composite loss function attains values higher than a set threshold. Figure 2 graphically describes such a composition.

Bayesian Optimization allows us to tackle this problem by first modeling the composite function $\mathrm{Loss}(s(\theta))$ as a Gaussian Process (GP) (Rasmussen, 2004). Modeling as a GP allows us to quantify

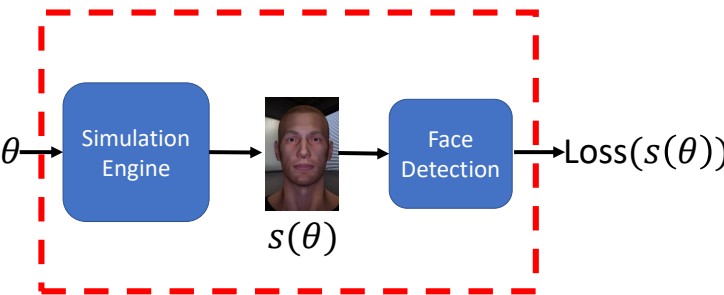

Figure 2: The composite function, which takes as input the simulation parameters $\theta$, in order to first produce a simulation $s(\theta)$. This in turn is fed into a machine classification system to compute a loss function characterizing whether the system performed correctly or poorly.

uncertainty around the predictions, which in turn is used to efficiently explore the parameter space in order to identify the spots that satisfy the search criterion. In this work, we follow the recommendations in (Snoek et al., 2012), and model the composite function (0-1 loss) via a GP with a Radial Basis Function (RBF) kernel, and use Expected Improvement (EI) as an acquisition function.

### 3.1 SIMULATED AGENT

Within the AirSim (Shah et al., 2018) environment we created an agent torso. The agent was placed in a room with multiple light sources (from windows, overhead lights and spotlights). Pictures of the avatar and the layout of the room are shown in Figure 1. The avatar and environment are being made available.

The skin type of the agent could be varied continuously from a lighter to darker skin type (allowing us to mimic different levels on the Fitzpatrick Classification Scale (Fitzpatrick, 1988)), and aging of the skin could be customized via the textures. The agent's facial position, facial actions (e.g., mouth opening or eye lids closing) could be fully customized.

In this paper, our goal was to illustrate how simulated data can be used to identify and evaluate biases within classifiers and not to exhaustively evaluate each face API's performance with respect to every facial expression or lighting configuration. However, our method could be used to do this.

### 3.2 PARAMETER SPACE

We manipulated the following parameters in order to evaluate how the appearance of the face impacted the success or failure of the face detection algorithms. The parameters were varied continuously within the bounds specified below. Angles are measured about the frontal position head position. For facial actions (mouth opening and eye lids closing) the mappings are specified in Facial Action Coding System (Ekman et al., 2002) intensities.

**Demographic Parameters**
- Skin Type: From lighter (Fitzpatrick I) to darker skin types (Fitzpatrick VI).
- Skin Age: From an unwrinkled complexion to a heavily wrinkled complexion.
**Head Pose and Expression Parameters:**
- Head Pitch: From -85 degrees (-1.5 radians) to 85 degrees (1.5 radians).
- Head Yaw: From -145 degrees (-2.5 radians) to 145 degrees (2.5 radians).
- Mouth Open: Mouth closed (FACS intensity 0) to open (FACS intensity 5).
- Eyes Closed: Eyes open (FACS intensity 0) to closed (FACS intensity 5).

Examples of differences in appearance with changes in these parameters are shown in Figure 3.

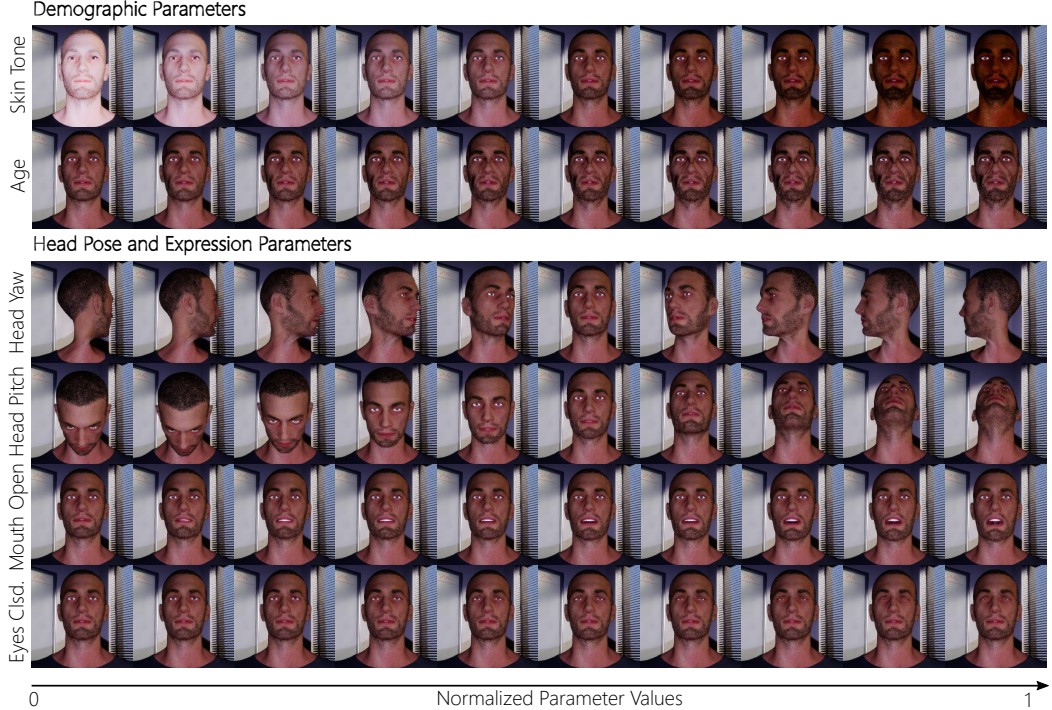

Figure 3: Top) Cropped views of the face from our simulated environment and variations in appearance with i) skin type, ii) age, iii) head yaw, iv) head pitch, v) mouth open, vi) eyes closed.

## 4    EXPERIMENTS AND RESULTS

We compared four facial analysis APIs in our experiments. Microsoft, Google, IBM and Face++ all offer services for face detection (and classification of other facial attributes). The APIs all accept HTTP POST requests with URLs of the images or binary image data as a parameter within the request. These APIs return JSON formatted data structures with the locations of the detected faces.

### 4.1    FACE DETECTION APIS

Unfortunately, detailed descriptions of the data used to train each of the face detection models were not available. To the best of our knowledge, the models exposed through these APIs all use deep convolutional neural network architectures and are trained on millions of images.

**Microsoft:** The documentation reports that a face is detectable when its size is $36 \times 36$ to $4096 \times 4096$ pixels and up to 64 faces can be returned for an image.

**Google:** The documentation did not report minimum size or resolution requirements.

**IBM:** The documentation reports that the minimum pixel density is $32 \times 32$ pixels per inch, and the maximum image size is 10 MB. IBM published a statement in response to the article by Buolamwini and Gebru (2017) in which they further characterize the performance of their face detection algorithm.[3]

**Face++:** The documentation reports that a face is detectable when its size is $48 \times 48$ to $4096 \times 4096$ pixels. The minimal height (or width) of a face should be also be no less than 1/48 of the short side of image. An unlimited number of faces can be returned per image.

---

[3]http://gendershades.org/docs/ibm.pdf

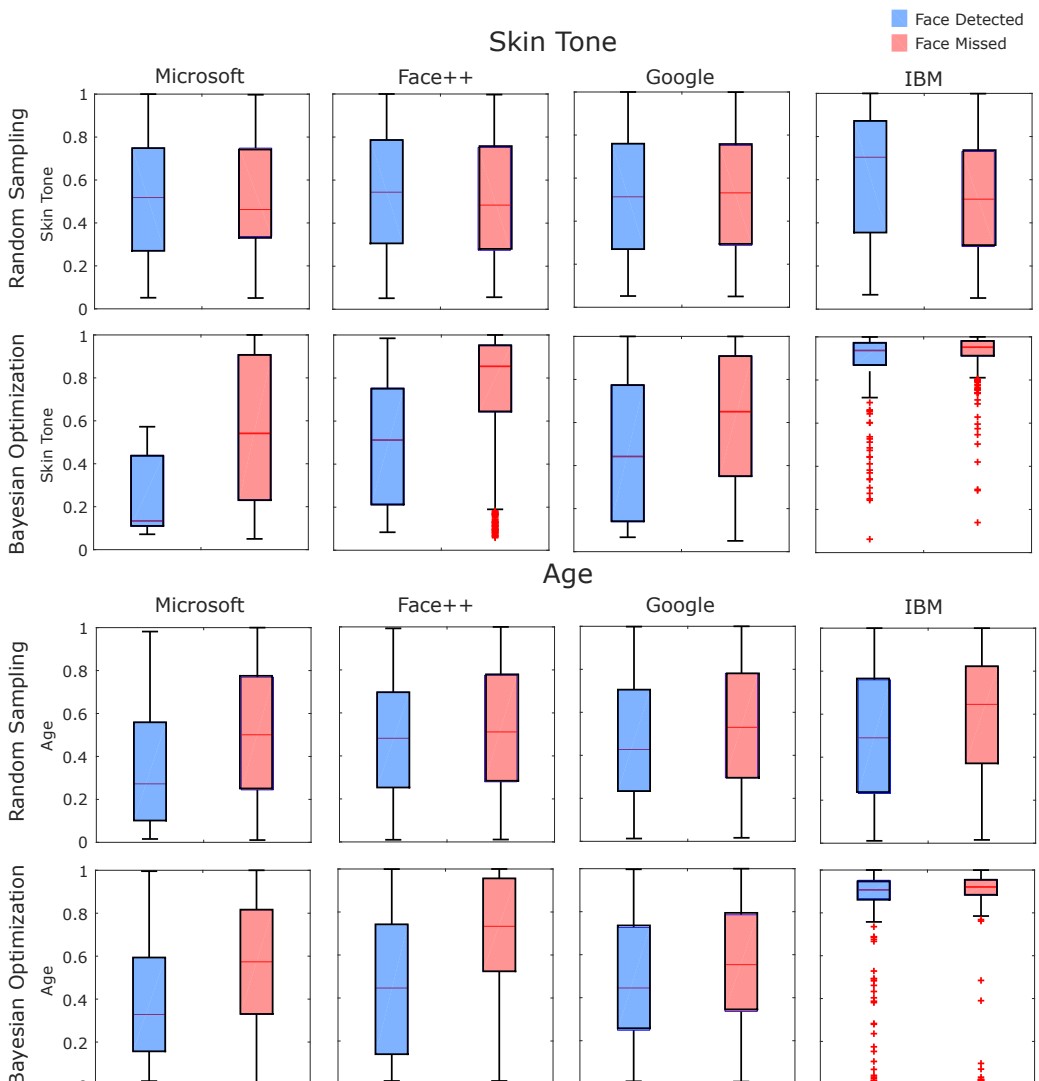

Figure 4: Distribution of each simulation parameter for successfully detected faces and missed detections. The distribution for skin types and ages skews darker and older respectively for false negatives than for true positives. The Bayesian optimization reveals this difference with fewer simulations, such that given 1000 samples the differences are apparent with BO and not with random sampling. The skin tone ranges from light (0) to dark (1) and age ranges from young (0) to old (1). For example images see Fig. 3

## 4.2 RESULTS

We used two approaches for searching our space of simulated faces for face detection failure cases. The first was randomly sampling parameters for generating face configurations and the second was using Bayesian optimization. We compared these sampling methods for the different face detection APIs. This type of analysis would have been difficult, if not impossible, without the ability to systematically synthesize data.

Figure 4 shows boxplots of the demographic parameters for faces correctly detected and faces missed. The results are shown for the skin type on a normalized scale of 0 (lighter skin type, Fitzpatrick I) to 1 (darker skin type, Fitzpatrick VI) and age on a normalized scale of 0 (unwrinkled) to 1 (heavily wrinkled). From left to right results illustrate the performance for the Microsoft, Face++, Google and IBM classifiers.

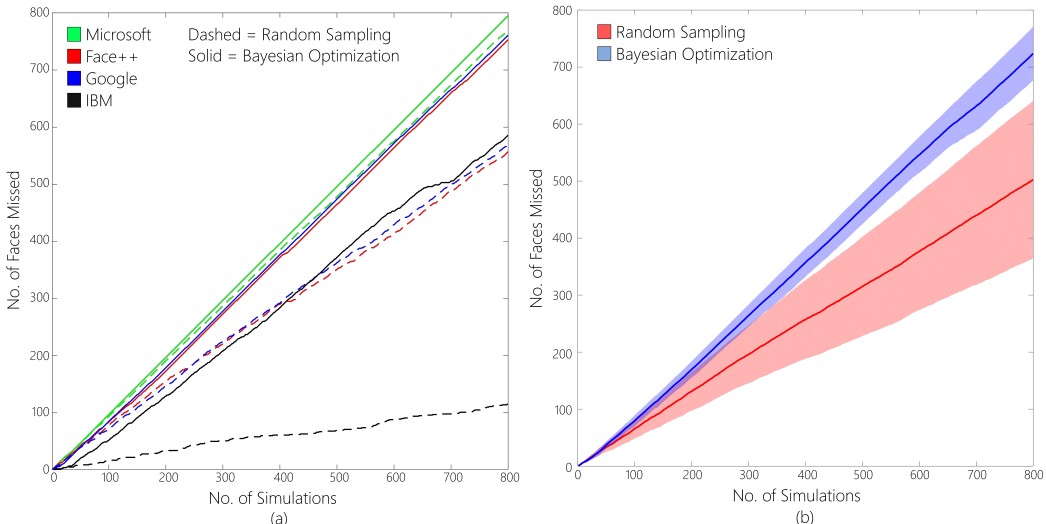

Figure 5: Sample efficiency of finding false negatives (missed face detections). a) Results for each of the face APIs individually. b) Results aggregated across all APIs. Red - Using random sampling (this is the same as chance performance). Blue - Using Bayesian optimization. Shaded area corresponds to the standard error.

Figure 5 shows the cumulative number of failure cases identified with increasing numbers of simulations (faces generated). The results are presented across all the APIs with the error bars representing the standard error. When using random sampling the sample (or simulation) efficiency for finding failure cases is significantly reduced compared to our Bayesian approach. After 800 samples 44% more failures (724 vs. 503) were found using Bayesian optimization.

## 5 DISCUSSION

### 5.1 PERFORMANCE ACROSS SKIN TYPE AND AGE

First, let us discuss the performance (characterized by true positives and false negatives) of the APIs with regard to the skin type and age appearance of the subjects. Figure 4 shows that the detectors consistently failed more frequently on faces with darker skin types and to a lesser extent on faces with older appearances. The missed detections were heavily skewed towards the upper end of the skin type and age ranges. Overall, the IBM API skew was the least extreme. This is probably due to the fact that the detectors were improved [4] following the results presented in (Buolamwini & Gebru, 2018) identifying biases within previous versions. This result shows that paying attention to the training data used to create the models behind these APIs can significantly mitigate the biases with the resulting models. While our results are only based on one base facial model, they provide results that support prior work Buolamwini (2017). While our results are only based on one base facial model, they provide evidence that supports prior work Buolamwini (2017). We would need further models with alternate bone structures to draw more conclusions about the generalizability of these results, especially with regard to other demographic variables (e.g., gender). The age manipulation of the avatar was somewhat constrained as only the texture (and not the shape) of the face was varied. This is a limitation and more realistic age manipulation warrants future work.

### 5.2 RANDOM VS. BAYESIAN SAMPLING

Using naive sampling techniques (e.g., random sampling or grid search), the cost of search is exponential with the number of parameters. Consider that there are three axes of head rotation, twenty-eight facial action units, thousands of possible facial expressions, and millions of potential lighting

---

[4]https://www.ibm.com/blogs/research/2018/02/mitigating-bias-ai-models/

configurations. Soon naive sampling techniques become impractical. We show that a Bayesian optimization approach can significantly reduce the number of simulations required to find an equal number of failure cases. In a simple experiment with six parameters, the Bayesian approach led to an over 40% improvement in efficiency with respect to finding the false negatives (missed detections). With a larger number of parameters this improvement will be much more dramatic.In addition, the improvement in sample efficiency was the most dramatic (over 500%) for the classifier with the fewest number of missed detections overall (IBM) (see Figure 5(a) black solid and dashed lines). This suggests that BO can further improve efficiency as failures become more challenging to find.

Bias in machine learned systems can be introduced in a number of ways. In complex models these biases can be difficult to identify. Using data generated via a realistic synthetic environment we have been able to identify demographic biases in a set of commercially available face detection classifiers. These biases are consistent with previous results on photo datasets (Buolamwini & Gebru, 2018). While the synthetic faces we used were not quite as realistic as photographs we believe that this empirical finding supports the use of parametric simulations for this problem.

## 6 Conclusions

We present an approach that leverages highly-realistic computer simulations to interrogate and diagnose biases within ML classifiers. We propose the use of simulated data and Bayesian optimization to intelligently search the parameter space. We have shown that it is possible to identify limits in commercial face detection systems using synthetic data. We highlight bias in these existing classifiers which indicates they perform poorly on darker skin types and on older skin texture appearances. Our approach is easily extensible, given the amount of parameters (e.g., facial expressions and actions, lighting direction and intensity, number of faces, occlusions, head pose, age, gender, skin type) that can be systematically varied with simulations.

We used one base facial model for our experimentation. This limits the generalization of our conclusions and the ability for us to determine whether the effects would be similar or different across genders and other demographic variables. Synthetic faces with alternate bone structures would need to be created to test these hypotheses. While the initial cost of creating the models is high, they can be used to generate large volumes of data, making synthetics cost effective in the long-run. Age modeling in face images should be improved using GAN or improved parametric synthetic models. A limitation of our work is that the aging was only represented via texture changes. We plan to investigate GAN-based approaches for synthesis and compare these to parametric synthesis. A hybrid of parametric and statistical models could be used to create a more controllable but diverse set of synthesized faces. Future work will consider retraining the models using synthetic data in order to examine whether this can be used to combat model bias.

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
