# OpenReview forum: "Identifying Bias in AI using Simulation"
_ICLR.cc/2019/Conference_

### Official Review · AnonReviewer3 · 2018-10-30
**Decent paper but I have several concerns.**

**Rating:** 6
**Confidence:** 2

**Review:**

This paper identifies bias of commercial Face detection API (Microsoft, Google, Face++, IBM) by sending face images generated from AirSim, in which different face attributes (e.g., skin color, age, face orientation, lighting conditions, etc) can be controlled and explored. The paper shows that for darker skin color and old age, the classifier tends to have a higher false negative rate (miss the face more). This is in particular more apparent if Bayesian Optimization is used to explore the parameter space based on the previous detection results to find the failure cases.

There are several concerns:

1. Bayesian Optimization might itself create a bias in the input data distribution, since it selectively pick some parameter configuration over the others.

2.  Using simulator might create additional biases. Maybe the faces generated by the simulator using the parameters of skin color of the minority / old age are less realistic than other faces, which lead to higher misclassification rate. In the paper there is no analysis in that aspect.

Overall I feel this is an interesting paper and it may identify important problems in the existing commercial AI system. However,  I am not an expert in this field so I am less confident about the thoroughness of experiments, as well as the fairness of approaches.

Minor issue:

Fig. 4 “Age”, skin detection => age.

---

> ### Author Response · Authors · 2018-11-09
> **Initial response and clarifications.**
>
> We are grateful for the review and questions. We are clarifying the following aspects in our paper and will upload a revised version in a couple of days.
>
> Bayesian optimization helps efficiently search for failure cases; however, the optimization does not effect which images the face detection model will fail upon. We included the results for the random parameter search to illustrate this.  If the BO had unfairly characterized the performance we would have expected the distributions of the face detection successes in the BO to be different from the random case. However, they are not significantly so.  We hope that this addresses your question.  If not, do let us know.  We will make sure this point is clear in the manuscript.
>
> We agree that the simulated faces are not identical to photographs.  However, our results corroborate previous findings about the biases in these commercial classifiers, see [1]. We believe this provides empirical evidence to help justify the use of simulation.
>
> [1] Buolamwini, J., & Gebru, T. (2018, January). Gender shades: Intersectional accuracy disparities in commercial gender classification. In Conference on Fairness, Accountability and Transparency (pp. 77-91).

---

### Official Review · AnonReviewer1 · 2018-11-02
**Important application area but findings are preliminary**

**Rating:** 7
**Confidence:** 5

**Review:**


Summary:
=========
The paper uses a proof-of-concept Bayesian parameter search-based simulation in virtual environment to probe biases of an already trained model towards specific categories that may have been sparsely represented in the training set. Understanding bias in trained models, especially in models involving end-to-end deep neural networks learners, is of high importance in machine learning. More specifically, probing the source of unintentional bias introduced as a result of skewed distribution in the training set and dissecting the biased performance is important for many applications such as surveillance, criminal profiling, medical diagnosis and predicting creditworthiness of a person.

The authors used four commercial face recognition APIs (by Microsoft, Google, IBM, and Face++) as test bed for this investigation.

Strength:
========
- The paper reads well and is easy to follow.

- The application of face detection and recognition is a good choice as it is precursor to detailed analysis in surveillance and criminal profiling.

- The choice of the controlled Bayesian parameter search enables one to control the amount of variation with respect to the expected uncertainty in performance of the classifier on the generated input from the simulator.

- The use of standardized measures such as Fitzpatrick’s skin tone and FACS intensity help in replication and consistency in the evaluation.

Weakness:
=========

- Although evaluating commercial APIs is good enough in demonstrating the existence of the bias, access to the trained model with possibility to retrain the model in such a way to mitigate a bias in particular parameter could have helped to further tie the drop in performance to the parameter variation.

- This work is preliminary and only involves a single person face manipulated in the parameter space. It lacks diversity of in samples and as such limits the analysis to draw strong conclusions. As such a generative network (such as a GAN trained to generate diverse samples while controlling for the parameters under investigation could have helped to draw more generalized conclusion.

- The age parameter variation is not convincingly different across the values considered. It would have been interesting to use models trained for age progression such as [1] for a more diverse variation in the age parameter space.

Minor comments:
===============
- Figures 4 and 5 could have used better captions describing the ranges for instance for age 1 is older and for skin tone 1 is darker (although indicated in Fig. 3). Captions should be self contained. Fig. 5 caption should describe the chance performance in each case.

- The manuscript should be revised to make in text citation formats consistent (some places it uses authors (year) and other places it uses (authors, year)). Also, minor punctuation and syntactic errors should be fixed.

[1] Yang, H., Huang, D., Wang, Y. and Jain, A.K., 2017. Learning face age progression: A pyramid architecture of gans. arXiv preprint arXiv:1711.10352.

---

> ### Author Response · Authors · 2018-11-09
> **Initial response and clarifications.**
>
> Thanks for the thoughtful review. We are updating the paper with the changes described below and will upload the revised version within the next few days.
>
> We agree that retraining classifiers with synthetic images is a very natural extension of this research. Our work aimed to specifically tackle the task of identifying biases. As we do not have access to all the commercially available API models we are not able to retrain and compare the performance at this point.  We feel that the performance of commercially available APIs is of particular interest.
>
> The reason we did not use GANs in this project but rather opted for a parametric model is that GANs are inherently statistically driven and therefore may well be subject to the same biases that the face detectors exhibit.  A parametric model; however, is less likely to suffer from these biases as it is not trained on data.  We are adding related work that uses synthetic data generated using GANs and some discussion related to this.
>
> e.g., Shrivastava, A., Pfister, T., Tuzel, O., Susskind, J., Wang, W., & Webb, R. (2017, July). Learning from Simulated and Unsupervised Images through Adversarial Training. In CVPR (Vol. 2, No. 4, p. 5).
>
> We are making the text citations consistent and are updating the captions to Figures 4 and 5.
>
> In Figure 5 chance performance is the same as random-sampling.  We are explicitly adding this to the caption to avoid confusion.

---

> > ### Comment · AnonReviewer1 · 2018-11-17
> > **A concern still remains**
> >
> > Thanks for the rebuttal.
> >
> > While I appreciate the authors making most of suggested changes, one major concern still remains. Under the 'Weakness' section of my review, bullet point #2 is not addressed well. My major concern is the simulation used only a single avatar. So, there is no diversity in facial features and other dimensional parameters whatsoever. Had the authors used dozen to hundreds of avatars, this concern could have been alleviated. Moreover, the despite the use of Bayesian optimizer to sample the parameters space, the pre-configured templates for each value of the parameters are not as convincing with respect to covering the parameter space. Especially reducing the age progression problem to mere increase and decrease of amount of a texture to the face is not realistic. That is why I mentioned GANs would have done a better job than the simulation, despite the ease of control in the virtual environment.
> >
> > I understand that GANs with vanilla encoder/decoders are themselves susceptible to bias. Effort could be made to minimize the said bias by:
> >
> > - carefully curating the training images for the GAN. I suspect that the authors can be careful in collecting their evaluation dataset as is the case with [1]
> >
> > - the authors could have also used parametrized encoder/decoder pair such as variational autoencoders within the GAN to carefully manipulate a particular parameter space. More specifically age progression has been studied well using GANs as I described in my original review.
> >
> > The use of a single avatar head as baseline severely limits any generalization of the evaluations. So, in the absence of such rigorous analysis and lack of sufficient representation of diversity in the simulation generated facial images, the authors need to water down their generalization. If that is possible, I am willing to increasing my rating depending on how well the paper discusses this specific limitation.
> >
> > [1] Buolamwini, J., & Gebru, T. (2018, January). Gender shades: Intersectional accuracy disparities in commercial gender classification. In Conference on Fairness, Accountability and Transparency (pp. 77-91).

---

> > > ### Author Response · Authors · 2018-11-19
> > > **Summary of additional revisions.**
> > >
> > > We really appreciate these concerns and wanted to emphasize that indeed we used one base facial bone structure (used to create the 3D model) in this work. Thus, we were not able to exhaustively test all possible facial appearances. We also agree that GANs are a good complementary approach that certainly warrants further exploration and there are controllable approaches for varying attribute values using GANs. Creating GANs that allow independent control over numerous facial attributes is promising.  We have changed the discussion and conclusions of our paper to temper the generalization of these results and suggest these alternatives. We have uploaded a revision of the paper with the following changes:
> > >
> > > Change to discussion section:
> > >
> > > While our results are only based on one base facial model, they provide evidence that supports prior work [1]. We would need further models with alternate bone structures to draw more conclusions about the generalizability of these results, especially with regard to other demographic variables (e.g., gender). The age manipulation of the avatar was somewhat constrained as only the texture (and not the shape) of the face was varied. This is a limitation and more realistic age manipulation warrants future work.
> > >
> > > Change to conclusions section:
> > >
> > > We used one base facial model for our experimentation. This limits the generalization of our conclusions and the ability for us to determine whether the effects would be similar or different across genders and other demographic variables. Synthetic faces with alternate bone structures would need to be created to test these hypotheses.  While the initial cost of creating the models is high, they can be used to generate large volumes of data, making synthetics cost effective in the long-run. Age modeling in face images should be improved using GAN or improved parametric synthetic models. A limitation of our work is that the aging was only represented via texture changes. We plan to investigate GAN-based approaches for synthesis and compare these to parametric synthesis.  A hybrid of parametric and statistical models could be used to create a more controllable but diverse set of synthesized faces. Future work will consider retraining the models using synthetic data in order to examine whether this can be used to combat model bias.
> > >
> > > [1] Buolamwini, J., & Gebru, T. (2018, January). Gender shades: Intersectional accuracy disparities in commercial gender classification. In Conference on Fairness, Accountability and Transparency (pp. 77-91).

---

### Official Review · AnonReviewer2 · 2018-11-04
**The paper uses computer graphics simulation for determining bias in commercial Face detection systems.  Specifically, skin type, age, head pose, expression parameters are varied to generate the data. Efficient exploration of simulation parameter space is done by using Bayesopt.**

**Rating:** 5
**Confidence:** 4

**Review:**

Quality and Clarity:  The paper is clear and has comprehensive references to the recent literature and past literature on face detection, bias in computer vision data and systems.
Originality:   Since the main claim of the paper is about the use of graphics simulation for performance characterization, we recommend that the authors review past work on use of simulations for systems performance characterization.  The idea of using simulations to perform performance assessment of vision goes back to the 90's (see for instance: haralick et al (haralick.org, performance characterization papers). The idea of using computer graphics simulations for transfer learning and performance assessment has been revisited recently (see for instance: veerasavarappu et al, 2015-17, arxiv papers, cvpr 2017, wacv 2017).

Significance:  While the paper demonstrates the utility of the main idea, the results are not comprehensive and can be strengthened.  For instance, the authors state that simulations can be used to combat bias via training with augmented data.   My opinion is that the paper may be more well suited in a applied workshop/conference such as WACV.

---

> ### Author Response · Authors · 2018-11-09
> **Initial response and clarifications**
>
> Thank you for the careful review and constructive comments. We are updating our manuscript and will upload a revised version in the next few days. We are grateful for you highlighting work that is relevant to our approach. We do agree that simulations have been used in the past for training and system performance evaluation.  We have added references and text related to the following work that we agree are important and relevant. However, none of these papers specifically addresses the issue of demographic bias in ML systems that we are aware of, which we feel is important enough an issue to warrant specific attention.  Furthermore, we propose to use Bayesian optimization to efficiently search the parameter space of such a parameterized model which helps reveal biases that would be more difficult to identify with naïve sampling of the parameter space.
>
> We have added the following references:
>
> Shrivastava, A., Pfister, T., Tuzel, O., Susskind, J., Wang, W., & Webb, R. (2017, July). Learning from Simulated and Unsupervised Images through Adversarial Training. In CVPR (Vol. 2, No. 4, p. 5).
>
> Veeravasarapu, V. S. R., Rothkopf, C., & Ramesh, V. (2016). Model-driven simulations for deep convolutional neural networks. arXiv preprint arXiv:1605.09582.
>
> Veeravasarapu, V. S. R., Hota, R. N., Rothkopf, C., & Visvanathan, R. (2015). Simulations for validation of vision systems. arXiv preprint arXiv:1512.01030.
>
> Veeravasarapu, V. S. R., Hota, R. N., Rothkopf, C., & Visvanathan, R. (2015). Model validation for vision systems via graphics simulation. arXiv preprint arXiv:1512.01401.
>
> Haralick, R., Performance characterization in computer vision. 60(2):245–249, September 1994.
>
> Vazquez, D., Lopez, A. M., Marin, J., Ponsa, D. and Geroimo. D., Virtual and real world adaptation for pedestrian detection. Pattern Analysis and Machine Intelligence, IEEE Transactions on, 36(4):797–809, 2014.

---

### Author Response · Authors · 2018-11-13
**Paper revision uploaded.**

We would like to thank the reviewers once again for all their thoughtful and helpful comments.  We have revised the manuscript, making the changes described in the "initial responses and clarifications".

---

### Comment · AnonReviewer2 · 2018-11-16
**Increased by rating by one level.**

Based on the revisions, I increase the paper rating by one level.

---

> ### Author Response · Authors · 2018-11-26
> **Thank you for your review.**
>
> We appreciate your reviews and the reconsideration of our manuscript.

---

### Meta-Review · Area_Chair1 · 2018-12-19
**Important problem, interesting solution, however stronger results may be needed**

**Recommendation:** Reject
**Confidence:** 3

**Metareview:**

 The paper addresses an important problem of detecting biases in classifiers (e.g. in face detection), using simulation tools with Bayesian parameter search. While the direction of research and the presented approach seem to be practically useful, there were several concerns raised by the reviewers regarding strengthening the results (e.g., beyond single avatar, etc), and suggestions on possibly a more applied conference as a better venue.  While thourough rebuttals by the authors addressed some of these concerns, which increased some ratings, overall, the paper was still in the borderline range. We hope the suggestions and comments of the reviewers can help to improve the paper.